# Mitochondrial DNA editing in mice with DddA-TALE fusion deaminases

Hyunji Lee[1,3], Seonghyun Lee[1,3], Gayoung Baek[1], Annie Kim[1,2], Beum-Chang Kang[1], Huiyun Seo[1] & Jin-Soo Kim [1,2✉]

DddA-derived cytosine base editors (DdCBEs), composed of the split interbacterial toxin DddA$_{tox}$, transcription activator-like effector (TALE), and uracil glycosylase inhibitor (UGI), enable targeted C-to-T base conversions in mitochondrial DNA (mtDNA). Here, we demonstrate highly efficient mtDNA editing in mouse embryos using custom-designed DdCBEs. We target the mitochondrial gene, *MT-ND5* (*ND5*), which encodes a subunit of NADH dehydrogenase that catalyzes NADH dehydration and electron transfer to ubiquinone, to obtain several mtDNA mutations, including m.G12918A associated with human mitochondrial diseases and m.C12336T that incorporates a premature stop codon, creating mitochondrial disease models in mice and demonstrating a potential for the treatment of mitochondrial disorders.

[1] Center for Genome Engineering, Institute for Basic Science, Daejeon, Republic of Korea. [2] Department of Chemistry, Seoul National University, Seoul, Republic of Korea. [3] These authors contributed equally: Hyunji Lee, Seonghyun Lee. ✉email: jskim01@snu.ac.kr

Mitochondrial DNA plays a critical role in cellular respiration via the mitochondrial oxidative phosphorylation (OXPHOS) system. Because the OXPHOS system is essential for survival, mutations in mtDNA cause severe malfunctions in multiple organs and muscles, especially in high-energy demand tissues[1]. Typically, in humans with a mitochondrial disease, wild-type (WT) and mutant mtDNA with single-base mutations coexist in a cell, resulting in a heteroplasmic state of the mtDNA population[2]. The balance between mutant and WT mtDNA determines the development of mitochondrial diseases with clinical phenotypes[3]. Programmable nucleases have been used to cleave a mutant mtDNA, but not the WT mtDNA to reduce the mutant mtDNA population in vitro and in vivo[4,5]. But these nucleases cannot induce or revert a specific mutation in mtDNA, possibly because DNA double-strand breaks are not efficiently repaired in mitochondria by nonhomologous end joining or homologous recombination, unlike those in the nucleus.

Mok et al. recently developed a base editing approach using the bacterial cytidine deaminase toxin, $DddA_{tox}$, to demonstrate efficient C-to-T base conversions in vitro[6]. In this approach, split $DddA_{tox}$ nontoxic halves fused to transcription activator-like effector (TALE) proteins, which can be custom-designed to recognize predetermined target DNA sequences[7], form a functional cytosine deaminase within the editing window to induce C-to-T base editing at the target site in mtDNA.

In this study, we investigate whether DdCBEs can achieve mtDNA base editing in vivo to create animal models with mitochondrial mutations and to show germline transmission of the resulting mitochondrial mutations in mice.

## Results

**Assembly of DdCBE plasmids**. To facilitate the assembly of custom-designed TALE arrays in DdCBEs, we constructed expression plasmids encoding split $DddA_{tox}$ halves and used our Golden-Gate cloning system, which employs a total of 424 (=6 × 64 tripartite + 2 × 16 bipartite + 2 × 4 monopartite) modular TALE array plasmids[8,9] (Fig. 1a). We mixed six TALE array plasmids and an expression vector in a single Eppendorf tube to construct a ready-to-use DdCBE plasmid that encodes 15.5–18.5 repeat variable diresidue arrays (Supplementary Fig. 1). The resulting DdCBEs recognized DNA sequences of 17–20 base pairs (bps) in length, including a conserved T at the 5′ terminus (Supplementary Table 1). As a result, a functional DdCBE pair recognized 32- to 40-bp DNA sequences (Fig. 1b).

**Mitochondrial DNA editing in vitro**. We chose the *Mus musculus* mitochondrial *ND5* gene encoding NADH-ubiquinone

oxidoreductase chain 5 protein to demonstrate in vivo mtDNA editing using our Golden-Gate assembly system. The ND5 protein is a core subunit of NADH dehydrogenase (ubiquinone), which catalyzes the transfer of electrons from NADH to the respiratory chain. In humans, mutations in the *ND5* gene are known to be related to mitochondrial encephalomyopathy, lactic acidosis, and stroke-like episodes (MELAS), as well as some symptoms of Leigh's syndrome and Leber's hereditary optic neuropathy (LHON)[10]. We sought to generate mouse models with genetic variations in the mitochondrial gene to mimic the dysfunctions in humans.

First, we assembled several DdCBE plasmids that were designed to induce two possible silent mutations, m.C12539T and m.G12542A, in the *ND5* gene. We transfected these plasmids into the NIH3T3 mouse cell line, and measured editing frequencies at day 3 post transfection. As expected, cytosine bases in the editing window were successfully converted into thymine with editing efficiencies of up to 19% (Fig. 2a). In line with the previous report showing that $DddA_{tox}$ exclusively deaminates cytosine in a "TC" motif[6], only the two cytosine bases in a TC context were edited. Indels and other types of point mutations were not detectably induced in the editing window.

**Mitochondrial DNA editing in vivo**. We chose the most active DdCBE pair (left-G1397-N and right-G1397-C) for an in vivo study. At day 4 post microinjection of in vitro transcripts encoding this DdCBE pair into one-cell stage C57BL6/J embryos, we were able to obtain nine edit-positive embryos out of a total of 32 embryos (28%; Table 1). The TALE–$DddA_{tox}$ deaminase efficiently induced C·G to T·A transitions, with frequencies that ranged from 2.2 to 25% at the m.C12539 position and from 0.63 to 5.8% at the m.G12542 position (Fig. 2b). Next, we implanted DdCBE-injected embryos into surrogate mothers, and obtained offspring carrying m.C12539T and m.G12542A mutations (Supplementary Fig. 2a). Three out of four newborn ($F_0$) mice harbored C·G to T·A conversions with frequencies that ranged from 1 to 27% (Fig. 2c). Two pups showed similar mutation frequencies in the toe and the tail, which were retained for at least 14 days after birth. Furthermore, these mtDNA mutations were detected in various tissues obtained from a fully grown adult $F_0$ mouse at day 50 post birth (Fig. 2d). This result suggests that mtDNA heteroplasmy induced by DdCBEs in one-cell stage zygotes can be maintained throughout development and differentiation.

To investigate whether the DdCBE-induced mutations can be transmitted to the next generation, we crossed the female $F_0$ mouse with a WT C57BL6/J male and obtained $F_1$ offspring. The m.C12539T and m.G12542A mutations were observed in two

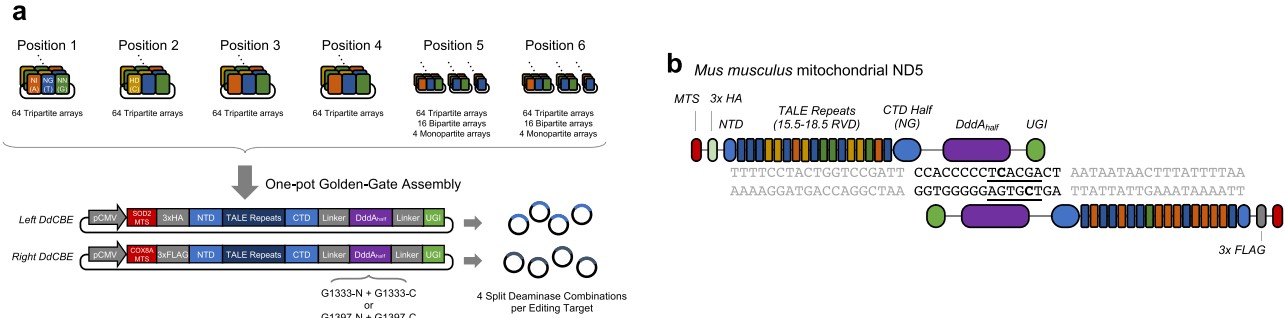

**Fig. 1 Schematic illustration for assembling DdCBE and its mitochondrial DNA editing. a** Scheme of one-pot Golden-Gate assembly for efficient DdCBE construction. A total of 424 arrays (64 tripartite arrays × 6 + 16 bipartite arrays × 2 + 4 monopartite arrays × 2) and expression vector were mixed to generate left and right modules for the final plasmid constructs. **b** Illustration of DdCBE interacting with mouse mitochondrial DNA target *ND5*. TALE-binding regions are shown in gray and base editing windows are depicted in black. Different repeat variable diresidue modules are shown in orange, blue, green, and yellow, which represent "NI" for adenine, "NG" for thymine, "NN" for guanine, and "HD" for cytosine recognition, respectively.

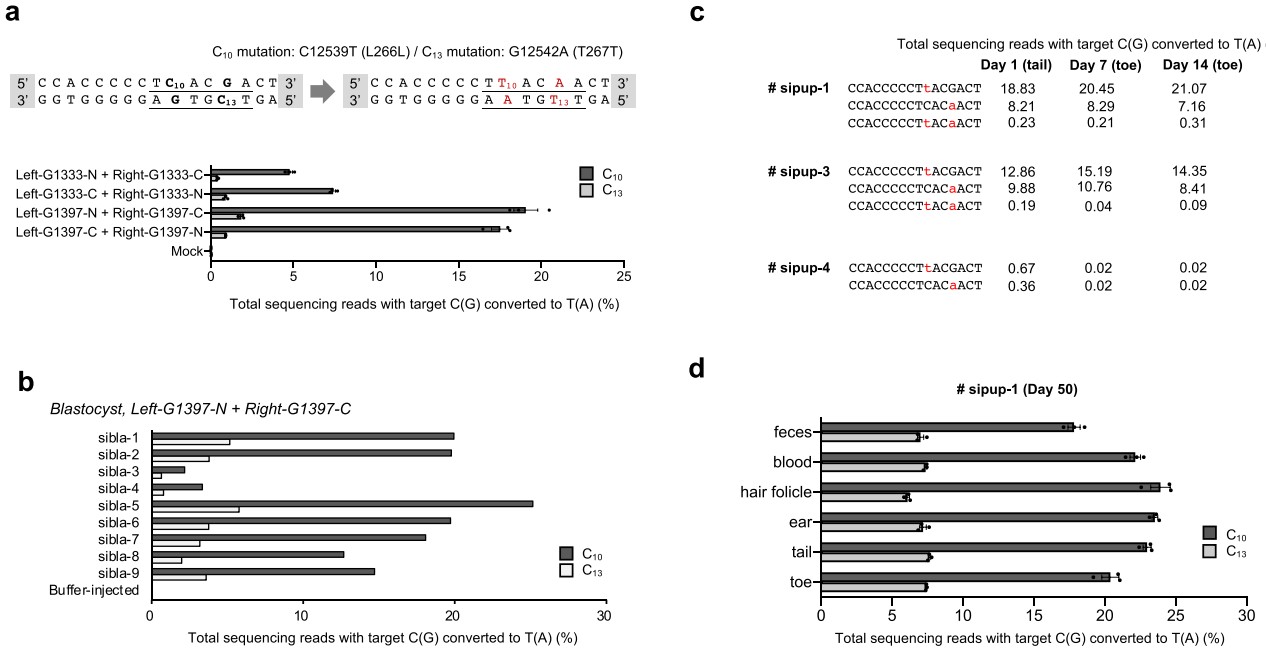

**Fig. 2 Mouse mitochondrial *ND5* point mutation generated by DdCBE-derived base editing. a** DdCBE deaminase-mediated cytosine-to-thymine base editing target and efficiency in NIH3T3 cells. In the target sequence, translation codons are underlined and possible editing loci are shown in red. Transfected combinations of DdCBE are annotated as left or right, -G1333 or -G1397, and -N or -C. *P* values of left-G1333-N + right-G1333-C, left-G1333-C + right-G1333-N, left-G1397-N + right-G1397-C, and left-G1397-C + right-G1397-N for $C_{10}$ mutation are 0.0012, 0.0003, 0.0014, and 0.0009, and for C13 mutation are 0.0116, 0.0076, 0.0030, and 0.0003, respectively (*$p < 0.05$ and **$p < 0.01$ using Student's two-tailed *t* test). **b** Corresponding base editing efficiency in mouse blastocysts. The sequencing data were obtained from blastocysts that developed after zygotes were microinjected with mRNA encoding the left-G1397-N and right-G1397-C DdCBE. **c** Alignments of mutant sequences from newborn pups. Targeted deep sequencing was performed using genomic DNA isolated from the tail of the newborns immediately after birth, and that from the toe 7 and 14 days after birth. Edited bases are shown in red. Editing frequencies in the mutant mitochondrial genome are shown. **d** Editing efficiencies in various tissues of an adult $F_0$ mouse (sipup-1). The sequencing data were obtained from each tissue 50 days after birth. In all graphs, the dark and light gray bars represent the frequency of m.C12539T ($C_{10}$) and m.G12542A ($C_{13}$) mutations, respectively. Error bars are the standard error of the mean (s.e.m.) for $n = 3$ biologically independent samples. Source data are provided in the Source data file.

**Table 1 Summary of the numbers of blastocysts used and mutants obtained.**

| Type of mutagenesis | Number of examined embryos | Number of blastocysts (%) | Number of transferred embryos | Number of offspring (%) | Number of edited/ total blastocysts (%) | Number of edited/ total offspring (%) |
|---|---|---|---|---|---|---|
| N/A (buffer injection) | 20 | 11 (55) | NA | NA | 0/11 (0) | NA |
| *ND5* silent | 56 | 32 (57) | 30 | 4 (13) | 9 (28) | 3 (75) |
| *ND5* G12918A | 79 | 44 (55) | 50 | 11 (22) | 11 (25) | 4 (36) |
| *ND5* STOP | 68 | 37 (54) | 120 | 27 (23) | 19 (51) | 9 (33) |

newborns with frequencies that ranged from 6 to 26% (Fig. 3a). Furthermore, these two mtDNA edits were detected at comparable frequencies in 11 different tissues (Fig. 3b).

**DdCBE-mediated *MT-ND5* G12918A mutation.** Next, we sought to produce the m.G12918A point mutation, which can cause mitochondrial disorders in humans. Note that mutations at this position give rise to multiple mitochondrial diseases, such as Leigh disease, MELAS syndrome, and LHON syndrome[10,11]. The cytosine base at this position has adjacent thymine, amenable for DdCBE-mediated base editing (Fig. 4a). We assembled four DdCBE pairs and confirmed successful base editing at this position in NIH3T3 cells with up to 6.4% efficiency (Fig. 4b). We then microinjected the best-performing DdCBE combination into mouse zygotes and measured editing efficiencies in the resulting blastocysts. Eleven out of 44 (25%) embryos harbored the m. G12918A mutation with editing efficiencies that ranged from 0.25

to 23% (Fig. 4c). Next, we implanted DdCBE-microinjected embryos into surrogate mothers to obtain the G12918A mutant offspring (Supplementary Fig. 2b). Four out of 11 newborn mice harbored the mutant allele with frequencies of 3.9–31.6% (Fig. 4d). Although no apparent phenotypes were observed right after birth, possibly because the pups were still very young and also because the WT mitochondrial DNA coexists with the mutant DNA in a heteroplasmic state, these data suggest that DdCBEs can be used to create animal models with mitochondrial disorders.

**MT-ND5 nonsense mutation.** Last but not the least, we investigated whether a loss-of-function mutation in *ND5* would be tolerated in mice by creating a nonsense mutation in the gene. We chose m.C12336 as a target cytosine for incorporation of a premature stop codon at the 199th position of the ND5 protein (Q199*; Fig. 5a). We first determined the editing activity of four custom-designed DdCBE combinations in transfected NIH3T3

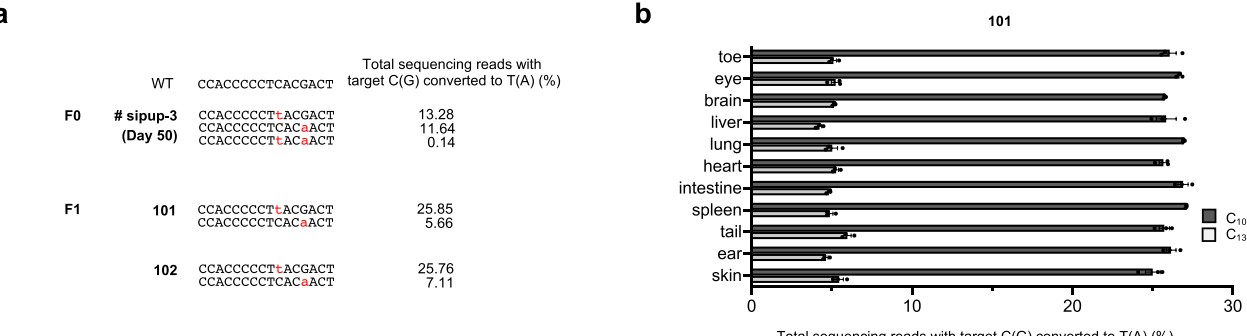

**Fig. 3 Germline transmission of mutant mtDNA. a** To observe the germline transmission of mtDNA mutations, the female $F_0$ (sipup-3) mouse was crossed with a wild-type C57BL6/J male to obtain $F_1$ pups (101, 102), after which targeted deep sequencing was performed. Edited bases are shown in red. Editing frequencies in the mutant mitochondrial genome are shown. **b** Base editing efficiencies in various tissues from an $F_1$ newborn pup (101), obtained using targeted deep sequencing of genomic DNA. Dark and light gray bars represent the frequency of m.C12539T ($C_{10}$) and m.G12542A ($C_{13}$) mutations, respectively. Error bars are the standard error of the mean (s.e.m.) for $n = 3$ biologically independent samples. Source data are provided in the Source data file.

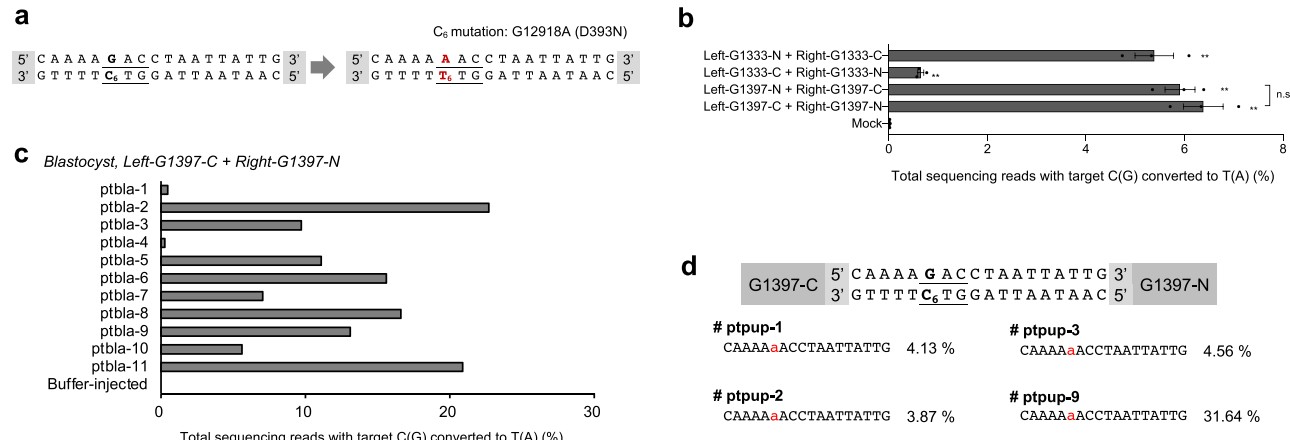

**Fig. 4 Mouse mitochondrial ND5 G12918A mutation induced by DdCBE. a** The DdCBE target for generating the m.G12918A point mutation, which would create a D393N change in the ND5 protein. The target codon is underlined and the possible editing locus is shown in red. **b** The efficiency of cytosine-to-thymine base editing with DdCBE in NIH3T3 cells. The annotations indicate the combination of DdCBE pairs that were co-transfected. Error bars are s.e.m. for $n = 3$ biologically independent samples (n.s. not significant, $^{*}p < 0.05$, and $^{**}p < 0.01$ using Student's two-tailed $t$ test). $P$ values of left-G1333-N + right-G1333-C, left-G1333-C + right-G1333-N, left-G1397-N + right-G1397-C, and left-G1397-C + right-G1397-N for $C_6$ mutation are 0.0052, 0.0099, 0.0027, and 0.0040, respectively. $P$ values for n.s. is 0.4971. **c** m.G12918A point mutation base editing efficiency in mouse blastocysts. The sequencing data were obtained from cultured blastocysts that developed after one-cell stage embryos were microinjected with mRNA encoding the left-G1397-C and right-G1397-N DdCBE. **d** Mice ($F_0$) carrying an ND5 point mutation. $F_0$ pups, which harbor an ND5 point mutation, that developed after microinjection of the DdCBE mRNAs. Corresponding alignment of mutant sequences from newborn pups. Edited bases are shown in red, and the column on the right indicates the editing frequencies in the mutant mitochondrial genome. Source data are provided in the Source data file.

cells and found that the most effective DdCBE pair induced the nonsense mutation at an editing efficiency of 5.7% (Fig. 5b). This DdCBE also induced a C-to-T conversion, albeit less efficiently, at m.G12341A in the editing window, which created a silent mutation (Q200Q). In 19 out of 37 (=51%) mouse embryos, we also observed the two m.C12336T and m.G12341A mutations with editing frequencies of up to 32% and 23%, respectively (Fig. 5c).

Encouraged by these results, we implanted mouse embryos into surrogate mothers, and obtained offspring with m.C12336T and m.G12341A mutations (Supplementary Fig. 2c). A total of 9 out of 27 $F_0$ mice (23%) harbored $C \cdot G$ to $T \cdot A$ conversions with frequencies that ranged from 0.22 to 57% (Fig. 5d, e), showing that the ND5 nonsense mutation did not lead to embryonic lethality.

## Discussion

Base editing, catalyzed by a Cas9 nickase fused to a deaminase protein, is a powerful method for inducing point mutations or substitutions in the nuclear genome without double-stranded DNA cleavage[12,13]. Previously, we were able to use both cytosine and adenine base editors to create animal models with point mutations in the nuclear genome and to correct them in vivo[14,15]. Unlike the nuclear genome, however, mitochondrial DNA has never been successfully edited in vitro or in vivo using the Cas9 nickase–deaminase fusion proteins, possibly because it is difficult to deliver both the protein component and guide RNA to mitochondria at the same time. Mok et al. demonstrated that a base editing system, free of guide RNA, composed of a TALE array fused to DddA_{tox} enables mtDNA base editing in cell lines. In this study, we used DdCBEs to create various point mutations, including silent, nonsense, and missense mutations, in the mitochondrial ND5 gene in mice for the first time (Table 1). To this end, we developed a Golden-Gate cloning system that consists of a total of 432 plasmids (8 expression plasmids plus 424 TALE array plasmids) to assemble DdCBE plasmids rapidly.

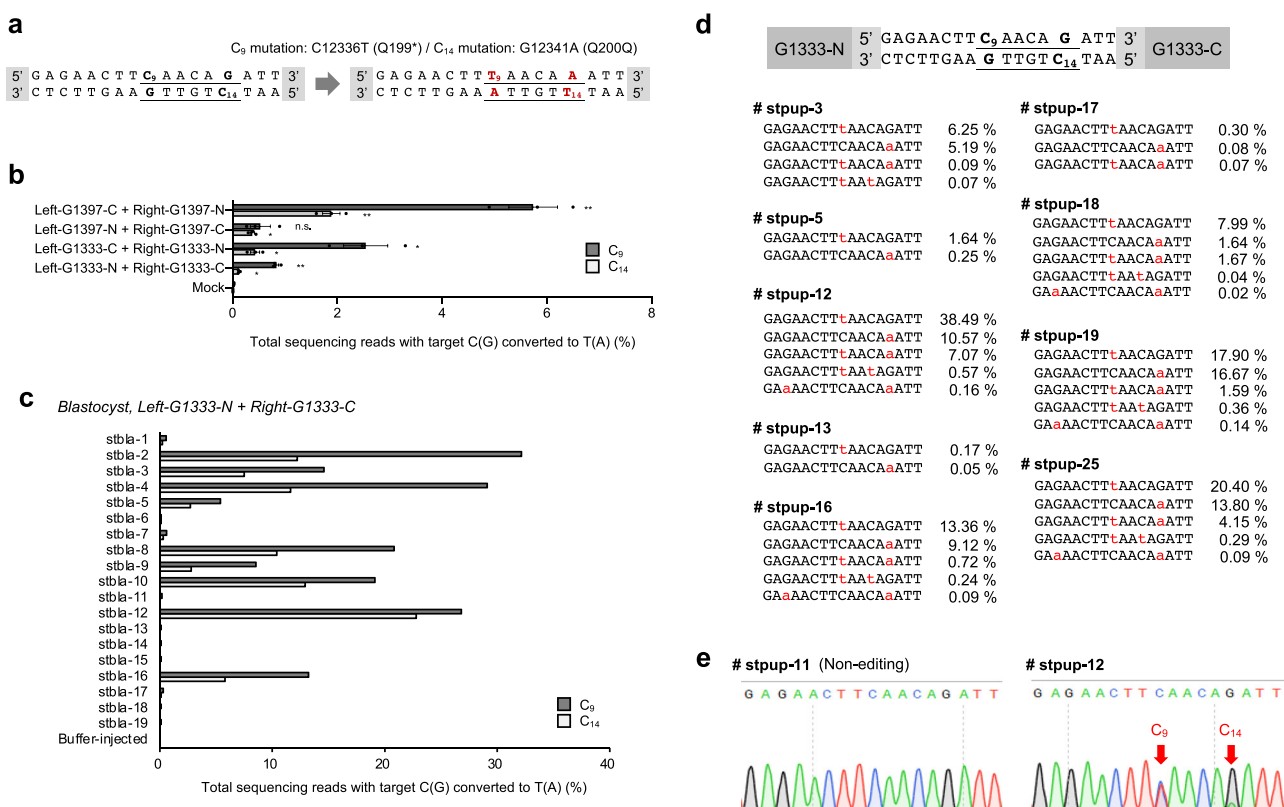

**Fig. 5 Mouse mitochondrial _ND5_ nonsense mutation generated via cytidine-deaminase-mediated base editing. a** The DdCBE target for generating the m. C12336T nonsense mutation and m.G12341A silent mutation. The m.C12336T ($C_9$) mutation creates a Q199stop mutation in the ND5 protein, whereas m. G12341A ($C_{14}$) causes a silent Q200Q mutation. Translation triplets are underlined and possible editing loci are shown in red. **b** The efficiency of the cytosine-to-thymine base editing that creates a nonsense mutation in NIH3T3 cells. The annotations indicate the combination of DdCBE pairs that were co-transfected into cells. Dark and light gray bars represent the frequency of m.C12336T ($C_9$) and m.G12341A ($C_{14}$) mutations, respectively. Error bars represent s.e.m. for $n = 3$ biologically independent samples (n.s. not significant, $^*p < 0.05$, and $^{**}p < 0.01$ using Student's two-tailed $t$ test). P values of left-G1333-N + right-G1333-C, left-G1333-C + right-G1333-N, left-G1397-N + right-G1397-C, and left-G1397-C + right-G1397-N for $C_9$ mutation are 0.0065, 0.1143, 0.0266, and 0.0037, and for $C_{14}$ mutation are 0.0077, 0.0144, 0.0406, and 0.0214, respectively. **c** Base editing efficiency in mouse blastocysts. The sequencing data were obtained from blastocysts that developed after zygotes were microinjected with mRNA encoding left-G1333-N and right-G1333-C DdCBE. Dark and light gray bars represent the frequency of the $C_9$ and $C_{14}$ mutations, respectively. **d** Alignment of mutant sequences from newborn pups. Edited bases are shown in red, and the column on the right indicates the editing frequencies in the mutant mitochondrial genome. **e** Sanger sequencing chromatograms from non-edited and edited mice. The red arrows indicates the substituted nucleotides. Source data are provided in the Source data file.

MtDNA mutations induced by the resulting DdCBEs were detected in various tissues in an adult mouse, showing that mtDNA heteroplasmy induced by DdCBEs was maintained throughout the development and differentiation. We also showed successful germline transmission of mtDNA edits induced by DdCBEs, suggesting that it is possible to create animal models with mitochondrial disorders. We propose that our Golden-Gate plasmids are valuable resources for studying the functions of mitochondrial genes in vitro and in vivo, and to correct pathogenic mutations for the treatment of mitochondrial genetic disorders in the future.

## Methods

**Plasmid construction**. We adapted our transcription activator-like effector nucleases (TALEN) assembly system to construct expression vectors for the split DddA halves, as well as final TALE–DddA$_{tox}$ constructs[8]. Beginning with the expression vector from the TALEN system, we replaced the fragments encoding the nuclear localization sequence and FokI obligatory heterodimeric halves with fragments encoding mitochondrial translocation sequences (MTS), DddA deaminase dimeric halves, and uracil glycosylase inhibitor (UGI). The MTS-, DddA-, and UGI-encoding sequences were synthesized by IDT. To construct expression plasmids, DNA fragments for Gibson assembly were amplified using Q5 DNA Polymerase (NEB), and subjected to PCR and gel purification. Purified gene fragments were assembled with a HiFi DNA

assembly kit (NEB); assembled plasmids were chemically transformed into _Escherichia coli_ DH5α (Enzynomics), and their identity confirmed by Sanger sequencing. Thereby, we obtained eight expression plasmids, which include BsaI restriction enzyme sites between regions encoding the N-terminal domain and C-terminal half domain (NG) of TALE for Golden-gate assembly. For DdCBE plasmid assembly, each expression plasmid was mixed with six module vectors (each encoding a TALE array), BsaI (10 U), and T4 DNA ligase (200 U), and reaction buffer in a single tube (Supplementary Fig. 1). Next, restriction–ligation reactions were performed in a thermocycler, with 20 cycles of 37 °C and 50 °C for 5 min each, followed by final incubations at 50 °C for 15 min and 80 °C for 5 min. Ligated plasmids were chemically transformed into _E. coli_ DH5α, and subjected to Sanger sequencing to confirm the identity of the constructs. Correct plasmids were midi-prepped (Qiagen) for cell transfection.

**Mammalian cell culture and transfection**. NIH3T3 (CRL-1658, American Type Culture Collection) cells were cultured and maintained at 37 °C with 5% CO$_2$. Cells were grown in DMEM supplemented with 10% (v/v) bovine calf serum (Gibco) without any antibiotics. For lipofection, cells were seeded in 12-well cell culture plates (SPL, Seoul, Korea) at a density of $1.5 \times 10^4$ cells per well, 18–24 h before transfection. Lipofection using Lipofectamine 3000 (Invitrogen) was performed with 500 ng of each TALE half monomer plasmid to make up 1000 ng of total plasmid DNA. Cells were harvested at day 3 post transfection.

**mRNA preparation**. The mRNA templates were prepared by PCR using Q5 High-Fidelity DNA Polymerase (NEB) with the following primers (F: 5′-CATCAA TGGGCGTGGATAG-3′, R: 5′-GACACCTACTCAGACAATGC-3′). DdCBE

mRNAs were synthesized using an in vitro RNA transcription kit (mMESSAGE mMACHINE T7 Ultra kit, Ambion) and purified with a MEGAclear kit (Ambion).

**Animals.** Experiments involving mice were approved by the Institutional Animal Care and Use Committee of Institute for Basic Science. Super ovulated C57BL/6 J females were mated to C57BL/6 J males, and females from the ICR strain were used as foster mothers. Mice were maintained in a specific pathogen-free facility under a 12 h dark–light cycle, and constant temperature (20–26 °C) and humidity maintenance (40–60%).

**Microinjection of mouse zygotes.** Steps prior to microinjection, including superovulation and embryo collection, as well as microinjection itself, were performed as described previously[16]. For microinjection, a mixture containing left DdCBE mRNA (300 ng/μl) and right DdCBE mRNA (300 ng/μl) was diluted in DEPC-treated injection buffer (0.25 mM EDTA, 10 mM Tris, pH 7.4), and injected into the cytoplasm of zygotes using a Nikon ECLIPSE Ti micromanipulator and a FemtoJet 4i microinjector (Eppendorf). After injection, embryos were cultured in micro drops of KSOM + AA (Millipore) at 37 °C for 4 days in a humidified atmosphere containing 5% $CO_2$. Two-cell-stage embryos were implanted into the oviducts of 0.5-d.p.c. pseudo-pregnant foster mothers.

**Genotyping.** Blastocyst stage embryos and tissues were incubated in lysis buffer (25 mM NaOH, 0.2 mM EDTA, pH 10) at 95 °C for 20 min, after which the pH was adjusted to 7.4 using HEPES (free acids, without pH adjustment) at a final concentration of 50 mM. Genomic DNA was extracted from pups for PCR genotyping using DNeasy Blood & Tissue Kits (Qiagen), and subjected to Sanger and targeted deep sequencing.

**Mitochondrial DNA isolation for high-throughput sequencing.** To isolate mitochondria from NIH3T3 cells in 12-well cell culture plates, the culture medium was aspirated, and 200 μl of Mitochondrial isolation buffer A (ScienCell) was added to each well. Cells were scraped with cell lifter, collected into microtubes, and homogenized with a disposable pestle designed for cell grinding. After 15 strokes, the homogenate was centrifuged at $1000 \times g$ for 5 min at 4 °C. The supernatant was transferred to a clean microtube and centrifuged at $10,000 \times g$ for 20 min at 4 °C. The pellet was resuspended in 20 μl of lysis solution (25 mM NaOH, 0.2 mM EDTA, pH 10), and incubated at 95 °C for 20 min. To lower the pH, we added 2 μl of 1 M HEPES (free acids, without pH adjustment) to the lysed mitochondrial solution. A total of 1 μl of lysate was used as a template for high-throughput sequencing.

**Targeted deep sequencing.** To create a high-throughput sequencing library, nested first PCR and second PCR were performed, and final index sequences were incorporated, using Q5 DNA Polymerase. The library was subjected to paired-end read sequencing using MiniSeq (Illumina). In all cases, the paired-end sequencing results were joined into a single fastqjoin file and analyzed via CRISPR RGEN Tools (http://www.rgenome.net/)[17].

**Data analysis and display.** Microsoft Excel (2019) and Powerpoint (2019) was used for drawing figures, graphs, and tables. Genome alignment, primer design, and cloning design were performed with Geneious (version 2021.0.1) and Snapgene 5.2.3, using NC_005089 genome as a reference.

**Reporting summary.** Further information on research design is available in the Nature Research Reporting Summary linked to this article.

## Data availability

The data that support the findings of this study are available from the corresponding author upon request. The high-throughput sequencing data from this study have been deposited in the NCBI Sequence Read Archive (SRA) database under the accession codes PRJNA694733 and PRJNA695094. Source data are provided with this paper.

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

## Acknowledgements

This work was supported by the Institute for Basic Science (IBS-R021-D1 to J.-S.K).

## Author contributions

H.L., S.L., and J.-S.K. designed the research. H.L., S.L., G.B., A.K., B.-C.K., and H.S. performed the experiments. J.-S.K. supervised the research. All authors discussed the results and commented on the manuscript.

## Competing interests

J.-S.K. is a cofounder of and holds stock in ToolGen. The other authors declare no competing interests.
