## [Peer Review File · Nature Communications]

Reviewers' Comments:

Reviewer #1:

Remarks to the Author:

The manuscript from Hyunji Lee and colleagues describes the application of the split DddA double strand DNA cytosine base editing system developed by the Liu laboratory for the introduction of different point mutations in mitochondrial genome encoded ND5 gene in mouse zygotes. The authors utilize a library of TALE modules in their Golden gate TALE assembly system to generate different TALE-split DddA heterodimers that recognize target loci for mitochondrial editing. mRNA delivery is utilized for these zygotic editing components, which results in appreciable mitoDNA editing both in treated blastocysts and treated F0 animals.

This study is the first to demonstrate successful zygotic editing of mitochondrial DNA using the split DddA system. The flexibility of this zygotic editing system is demonstrated through the successful generation of edits at three different target sites. The generation of viable animals with moderate levels of mitoDNA editing using this approach will be of interest to the scientific community, as it provides a potential vehicle for developing animal models of different mitochondrial disorders. The adaptation of the golden gate system for TALE assembly from the library of sub-assembled modules will provide the scientific community with the capability of leveraging the TALE-split DddA editors for a variety of purposes.

The major question that the current manuscript leaves unanswered is if the mitochondrial mutations can be passed on through the female germline. For this mitochondrial editing system to provide a robust method for studying the in vivo consequences of different mitochondrial mutations, it would be important to establish the ability to create mouse strains carrying the mitoDNA mutations of interest. The addition of these data would demonstrate the feasibility of this approach.

Minor improvements:

- One of the strengths of this manuscript is the incorporation of a facile golden gate TALE assembly system. It would be valuable to the scientific community if the authors could share their expertise in the form of more details in the methods on the choice of the length of TALE array to assemble for a target site, or at least a statement of reference where additional information on the choice and length of TALE module vectors can be found so that readers will have some guidance for their own assemblies. It is not immediately obvious what lengths of TALE arrays were used for targeting m.G12918A or m.C12336T. For Figure 1a or b it would also be useful to note in the legend what the module color scheme represents with regards to base recognition. In addition, a plasmid map or schematic layout of the eight expression plasmids with BsaI sites in a supplementary figure would be valuable for reference for those readers that are new to Golden gate assembly.

- One additional discussion point that could be included in the manuscript is information on the presence of other types of mutations (indels) at the target sites. The authors do not describe other events besides base transitions in the target window. It would be valuable to comment on the absence of other mutagenic events if this is what they have observed.

Reviewer #2:

Remarks to the Author:

The manuscript by Hyunji Lee and colleagues describe the use of a newly developed approach from the Liu's Lab to base-edit mtDNA. The original work was done in cultured cells. The novelty of this paper is that they quickly adapted it in embryos and found that they could base edit mtDNA and have pups with heteroplasmic mtDNA mutations, including the m.G12918A associated with human mitochondrial diseases and the m.C12336T that incorporates a premature stop codon in

ND5. Efficiency of base-editing was comparable to what the Liu's lab reported, and maybe even better in embryos when compared to the same base editing in cultured cells.

The authors emphasized, as a major point, that their golden-gate system makes it easy to assemble the TALE domain, which is useful, but that approach was well documented for TALENs already.

The work is straightforward and the advance is important. I really have no major issues with the experiments or the message. There are a few pieces of information missing though. Are the mtDNA mutations obtained by base-editing embryos transmitted to the F1 progeny. I understand that the authors are rushing to be the first to use this approach in mice, but it is a critical piece of information, in my opinion. Also, they only analyzed the tail of newborn and toes of 1 week old mice. Are these animals dead? It would be interesting to see tissue distribution of mtDNA heteroplasmy in F0 and F1.

POINT-BY-POINT RESPONSE

We would like to thank reviewers for helpful comments. The two reviewers suggested that we should show germline transmission of the mtDNA mutations in our revised manuscript. We have now included new data showing germline transmission and revised our manuscript as shown below in detail. In addition, we have divided the main text into four sections including Introduction, Results, Discussion, and Methods. Added or edited sentences are underlined in the main text for clarity.

Reviewer #1 (Remarks to the Author):

This study is the first to demonstrate successful zygotic editing of mitochondrial DNA using the split DddA system. The flexibility of this zygotic editing system is demonstrated through the successful generation of edits at three different target sites. The generation of viable animals with moderate levels of mitoDNA editing using this approach will be of interest to the scientific community, as it provides a potential vehicle for developing animal models of different mitochondrial disorders. The adaptation of the golden gate system for TALE assembly from the library of sub-assembled modules will provide the scientific community with the capability of leveraging the TALE-split DddA editors for a variety of purposes.

The major question that the current manuscript leaves unanswered is if the mitochondrial mutations can be passed on through the female germline. For this mitochondrial editing system to provide a robust method for studying the in vivo consequences of different mitochondrial mutations, it would be important to establish the ability to create mouse strains carrying the mitoDNA mutations of interest. The addition of these data would demonstrate the feasibility of this approach.

Response: To confirm germline transmission of mtDNA edits, we obtained two F1 mice by crossing a mutant F0 mouse with a wild-type mouse. We have now added the following sentences in our revised manuscript and corresponding data in Figure 3a.

On Page 6,

To investigate whether the DdCBE-induced mutations can be transmitted to the next generation, we crossed the female F₀ mouse with a wild-type C57BL6/J male and obtained F₁ offspring. The m.C12539T and m.G12542A mutations were observed in two newborns with frequencies that ranged from 6% to 26% (Fig. 3a). Furthermore, these two mtDNA edits were detected at comparable frequencies in 11 different tissues (Fig. 3b).

a

	WT	CCACCCCTCAGACT	Total sequencing reads with target C(G) converted to T(A) (%)
F0	# sipup-3 (Day 50)	CCACCCCTtACGACT	13.28
		CCACCCCTCACaACT	11.64
		CCACCCCTtACaACT	0.14
F1	101	CCACCCCTtACGACT	25.85
		CCACCCCTCACaACT	5.66
F1	102	CCACCCCTtACGACT	25.76
		CCACCCCTCACaACT	7.11

Figure 3.

Germline transmission of mutant mtDNA. (a) To observe the germline transmission of mtDNA mutations, the female F₀ (sipup-3) mouse was crossed with a wild-type C57BL6/J male to obtain F₁ pups (101, 102), after which targeted deep sequencing was performed. Edited bases are shown in red. Editing frequencies in the mutant mitochondrial genome are shown. (b) Base editing efficiencies in various tissues from an F₁ newborn pup (101), obtained using targeted deep sequencing of genomic DNA. Dark and light gray bars represent the frequency of m.C12539T (C₁₀) and m.G12542A (C₁₃) mutations, respectively. Error bars are the standard error of the mean (s.e.m.) for n = 3.

Minor improvements:

•One of the strengths of this manuscript is the incorporation of a facile golden gate TALE assembly system. It would be valuable to the scientific community if the authors could share their expertise in the form of more details in the methods on the choice of the length of TALE array to assemble for a target site, or at least a statement of reference where additional information on the choice and length of TALE module vectors can be found so that readers will have some guidance for their own assemblies. It is not immediately obvious what lengths of TALE arrays were used for targeting m.G12918A or m.C12336T.

Response: We have now included additional information in the main text and in Supplementary Information as shown below.

On Page 4,

Assembly of DdCBE plasmids

To facilitate the assembly of custom-designed TALE arrays in DdCBEs, we constructed expression plasmids encoding split DddA_{tox} halves and used our Golden-Gate cloning system, which employs a total of 424 (= 6 x 64 tripartite + 2 x 16 bipartite + 2 x 4 monopartite) modular TALE array plasmids⁹ (Fig. 1a). We mixed six TALE array plasmids and an expression vector in a single Eppendorf tube to construct a ready-to-use DdCBE plasmid that encodes 15.5-18.5 repeat variable diresidue (RVD) arrays (Supplementary Fig. S1). The resulting DdCBEs recognized DNA sequences of 16-20 base pairs (bps) in length, including a conserved T at the 5'

terminus (**Supplementary Table 1**). As a result, a functional DdCBE pair recognized 32- to 40-bp DNA sequences (**Fig. 1b**).

Supplementary Figure 1. Scheme of Golden-Gate cloning for generating the DdCBE construct. All reactions simultaneously occur in a single tube; the arrows do not indicate a sequential reaction process. The Bsal enzyme was used to cut the empty expression vector and the module vector, leaving the linearized backbone and TALE module inserts with compatible cohesive end. T4 DNA ligase then ligated the backbone and six module inserts to create the final DdCBE construct. Eight DdCBE cloning backbone plasmids were used in this study: Left-G1333-N, Left-G1333-C, Left-G1397-N, and Left-G1397-C for SOD2 MTS; Right-G1333-N, Right-G1333-C, Right-G1397-N, and Right-G1397-C for COX8A MTS.

Supplementary Table 1. TALE arrays used in this study.

	Left TALE Target Sequence	Right TALE Target Sequence
ND5 Silent Mutation	5'-T TTTCTACTGGTCCGAT T-3'	5'-T TAAAATAAAGTTATTAT T-3'
ND5 G12918A Mutation	5'-T TGCAGGTATTAATTGCT T-3'	5'-T TCCTAACAGGGTTCTAC T-3'
ND5 Nonsense Mutation	5'-T TCCCTAACATAAACTCA T-3'	5'-T TGTTGTTGGAGAATA T-3'

We chose TALE arrays with effective DNA binding according to results in our previous report (Kim *et al.*, *Nat. Biotech.*, 31.3 (2013): 251-258). The first and last thymine residues are essential for recognition of target base pairs in our system. The designed TALEs can recognize DNA sequences that are 17–20 bps in length, including a conserved T at the 5' end.

For Figure 1a or b it would also be useful to note in the legend what the module color scheme represents with regards to base recognition.

Response: We have now added the description in the figure legend, as shown below.

Figure 1.

(a) Scheme of one-pot Golden-Gate assembly for efficient DdCBE construction. A total of 424 arrays (64 tripartite arrays \times 6 + 16 bipartite arrays \times 2 + 4 monopartite arrays \times 2) and expression vector were mixed to generate left and right modules for the final plasmid constructs. (b) Illustration of DdCBE interacting with mouse mitochondrial DNA target ND5. TALE-binding regions are shown in gray and base editing windows are depicted in black. Different repeat variable diresidue modules are shown in orange, blue, green, and yellow, which represent 'NI' for adenine, 'NG' for thymine, 'NN' for guanine, and 'HD' for cytosine recognition, respectively.

In addition, a plasmid map or schematic layout of the eight expression plasmids with Bsal sites in a supplementary figure would be valuable for reference for those readers that are new to Golden gate assembly.

Response: The schematic layout of the expression plasmids is shown in Supplementary Fig. 1.

Supplementary Figure 1. Scheme of Golden-Gate cloning for generating the DdCBE constructs. All reactions simultaneously occur in a single tube; the arrows do not indicate a sequential reaction process. The Bsal enzyme was used to cut the empty expression vector and the module vector, leaving the linearized backbone and TALE module inserts with compatible cohesive end. T4 DNA ligase then ligated the backbone and six module inserts to create the final DdCBE construct. Eight DdCBE cloning backbone plasmids were used in this study: Left-G1333-N, Left-G1333-C, Left-G1397-N, and Left-G1397-C for SOD2 MTS; Right-G1333-N, Right-G1333-C, Right-G1397-N, and Right-G1397-C for COX8A MTS.

• One additional discussion point that could be included in the manuscript is information on the presence of other types of mutations (indels) at the target sites. The authors do not describe other events besides base transitions in the target window. It would be valuable to comment on the absence of other mutagenic events if this is what they have observed.

Response: To address this important issue, we have now added a sentence in the main text, as shown below.

On Page 5,

... In line with the previous report showing that DddA_{tox} exclusively deaminates cytosine in a ‘TC’ motif⁷, only the two cytosine bases in a TC context were edited. Indels and other types of point mutations were not detectably induced in the editing window.

Reviewer #2 (Remarks to the Author):

The work is straightforward and the advance is important. I really have no major issues with the experiments or the message. There are a few pieces of information missing though. Are the mtDNA mutations obtained by base-editing embryos transmitted to the F1 progeny.

Response: To address this issue, we obtained two F1 mice by crossing a mutant F0 mouse with an wild-type mouse. We have now added the following sentences in our revised manuscript and corresponding data in Figure 3a.

On Page 6,

To investigate whether the DdCBE-induced mutations can be transmitted to the next generation, we crossed the female F₀ mouse with a wild-type C57BL6/J male and obtained F₁ offspring. The m.C12539T and m.G12542A mutations were observed in two newborns with frequencies that ranged from 6 % to 26 % (Fig. 3a). Furthermore, these two mtDNA edits were detected at comparable frequencies in 11 different tissues (Fig. 3b).

a

	WT	CCACCCCTCAGACT	Total sequencing reads with target C(G) converted to T(A) (%)
F0	# sipup-3 (Day 50)	CCACCCCT t ACGACT	13.28
		CCACCCCTCAC a ACT	11.64
		CCACCCCT t AC a ACT	0.14
F1	101	CCACCCCT t ACGACT	25.85
		CCACCCCTCAC a ACT	5.66
F1	102	CCACCCCT t ACGACT	25.76
		CCACCCCTCAC a ACT	7.11

Figure 3.

Germline transmission of mutant mtDNA. (a) To observe the germline transmission of mtDNA mutations, the female F₀ (sipup-3) mouse was crossed with a wild-type C57BL6/J male to obtain F₁ pups (101, 102), after which targeted deep sequencing was performed. Edited bases are shown in red. Editing frequencies in the mutant mitochondrial genome are shown. (b) Base editing efficiencies in various tissues from an F₁ newborn pup (101), obtained using targeted deep sequencing of genomic DNA. Dark and light gray bars represent the frequency of m.C12539T (C₁₀) and m.G12542A (C₁₃) mutations, respectively. Error bars are the standard error of the mean (s.e.m.) for n = 3.

I understand that the authors are rushing to be the first to use this approach in mice, but it is a critical piece of information, in my opinion. Also, they only analyzed the tail of newborn and toes of 1 week old mice. Are these animals dead? It would be interesting to see tissue distribution of mtDNA heteroplasmy in F₀ and F₁.

Response: We have now added new data showing tissue distribution of the mtDNA edits in F₀ and F₁ mice and described the results in the main text, as shown below.

Figure 2. Mouse mitochondrial ND5 point mutation generated by DdCBE-derived base editing. ... (d) Editing efficiencies in various tissues of an adult F₀ mouse (sipup-1). The sequencing data were obtained from each tissue 50 days after birth. In all graphs, the dark and light gray bars represent the frequency of m.C12539T (C₁₀) and m.G12542A (C₁₃) mutations, respectively. Error bars are the standard error of the mean (s.e.m.) for n = 3 ($p < 0.05$ and $p < 0.01$ using Student's two-tailed *t*-test).

On Page 5,
Two pups showed similar mutation frequencies in the toe and the tail, which were retained for at least 14 days after birth. Furthermore, these mtDNA mutations were detected in various tissues obtained from a fully-grown adult F₀ mouse at day 50 post-birth (Fig. 2d). This result suggests that mtDNA heteroplasmy induced by DdCBEs in one-cell stage zygotes can be maintained throughout development and differentiation.

Figure 3.

Germline transmission of mutant mtDNA. (a) To observe the germline transmission of mtDNA mutations, the female F₀ (sipup-3) mouse was crossed with a wild-type C57BL6/J male to obtain F₁ pups (101, 102), after which targeted deep sequencing was performed. Edited bases are shown in red. Editing frequencies in the mutant mitochondrial genome are shown. (b) Base editing efficiencies in various tissues from an F₁ newborn pup (101), obtained using targeted deep sequencing of genomic DNA. Dark and light gray bars represent the frequency of m.C12539T (C₁₀) and m.G12542A (C₁₃) mutations, respectively. Error bars are the standard error of the mean (s.e.m.) for $n = 3$.

On Page 6,

To investigate whether the DdCBE-induced mutations can be transmitted to the next generation, we crossed the female F₀ mouse with a wild-type C57BL6/J male and obtained F₁ offspring. The m.C12539T and m.G12542A mutations were observed in two newborns with frequencies that ranged from 6% to 26% (Fig. 3a). Furthermore, these two mtDNA edits were detected at comparable frequencies in 11 different tissues (Fig. 3b).

Reviewers' Comments:

Reviewer #1:

Remarks to the Author:

The authors have addressed my minor concerns, and have importantly demonstrated germline transmission of the mtDNA mutations to the F1 generation.

This manuscript will provide an important blueprint for other laboratories that are interested in employing the split deaminase system for the construction of animal models with mtDNA mutations.